# Impact of Forest Harvesting Intensity and Water Table on Biodegradability of Dissolved Organic Carbon in Boreal Peat in an Incubation Experiment

Elina Peltomaa [1,*] , Mari Könönen [2], Marjo Palviainen [1], Annamari (Ari) Laurén [2] , Xudan Zhu [3],
Niko Kinnunen [4] , Heidi Aaltonen [5], Anne Ojala [6] and Jukka Pumpanen [4]

1   Department of Forest Sciences, University of Helsinki, 00014 Helsinki, Finland; marjo.palviainen@helsinki.fi
2   Faculty of Science and Forestry, University of Eastern Finland, 80101 Joensuu, Finland;
    mari.kononen@uef.fi (M.K.); ari.lauren@uef.fi (A.L.)
3   Department of Environmental and Biological Sciences, University of Eastern Finland, 80101 Joensuu, Finland;
    xudan.zhu@uef.fi
4   Department of Environmental and Biological Sciences, University of Eastern Finland, 70211 Kuopio, Finland;
    niko.kinnunen1@uef.fi (N.K.); jukka.pumpanen@uef.fi (J.P.)
5   Department of Agricultural Sciences, University of Helsinki, 00014 Helsinki, Finland;
    heidi.m.aaltonen@helsinki.fi
6   Bioeconomy and Environment, Natural Resources Institute Finland, 00790 Helsinki, Finland;
    anne.ojala@luke.fi
*   Correspondence: elina.peltomaa@helsinki.fi

**Abstract:** Boreal peatlands are vast carbon (C) stores but also major sources of dissolved organic C (DOC) and nutrients to surface waters. Drainage and forest harvesting accelerates DOC leaching. Continuous cover forestry (CCF) is considered to cause fewer adverse environmental effects. Yet, the effects of CCF on DOC processes are unrecognised. We study DOC production and quality in unharvested, CCF, and clear-cut drained peatland forests and in a non-forested alluvial sedge fen. Parallel replicate peat columns with ground vegetation are collected from the uppermost 50 cm at each site, and the water table (WT) is set to −20 or −40 cm depths on the columns. During the eight-month ex situ incubation experiment, the soil water samples are extracted monthly or bi-monthly. The samples are incubated at 15 °C for multiple 72 h incubation cycles to study pore water quality and biodegradation of DOC. The $CO_2$ production occurs during the first three days. The DOC concentrations and the $CO_2$ release per volume of water are significantly lower in the sedge fen than in the drained peatland forests. The WT has a negligible effect on DOC concentrations and no effect on DOC quality, but the higher WT has generally higher $CO_2$ production per DOC than the lower WT. The results suggest that peat in the drained peatlands is not vulnerable to changes per se but that forest management alters biotic and abiotic factors that control the production, transport, and biodegradation of DOC.

**Keywords:** peatland forestry; continuous cover forestry; clear-cutting; dissolved organic carbon (DOC); biodegradability; water quality; drainage

## 1. Introduction

Peatlands store one-third of the global soil carbon (C) as they contain 545 Gt to 1055 Gt of C [1–4]. Most of the peatlands are located in boreal regions, where the cold temperature and high water table (WT) maintain a slow rate of organic matter decay, supporting the accumulation of peat [5]. The WT also exerts control over the lateral transport of C from peatlands to watercourses. This dissolved organic carbon (DOC) flux represents 10–20% of the total C release from peatlands, being an important component of regional C balance [6,7].

Leaching of DOC enhances the surface water browning and affects the release of the greenhouse gases (GHGs) in the receiving aquatic systems as DOC biologically and photochemically degrades to carbon dioxide ($CO_2$) and methane ($CH_4$) [8,9]. The DOC export is sensitive to climate change and anthropogenic disturbance, and especially in managed peatlands it can turn the peatlands from net C sinks to net C sources [9,10]. Peatland forest management practices, such as harvesting and drainage, cause decades of elevated nutrient and DOC exports to downstream watercourses [11–14]. Therefore, the environmental viability of the harvesting and drainage of forested peatlands has been recently questioned.

The DOC includes compounds with various molecular weights (MWs) and biodegradability. The labile (low-MW; i.e., sugars and aliphatic C chains) DOC is primarily from plant exudates and leachates, while the more recalcitrant (high-MW; aromatic ring structures) DOC is microbial-derived originating from decomposition processes [15,16]. The biodegradability of DOC plays a key role in its lateral transportation: the labile fraction is degraded in a few days, whereas the half-life of the recalcitrant DOC ranges from months to years [15]. The DOC in peatlands is typically high in humic, fulvic, phenolic, and uronic acids, having high MW and low biodegradability, which increases the lifetime and the risk of DOC transport into watercourses [15,17].

The primary factors regulating organic matter decomposition, and thus the production of recalcitrant DOC in peatlands, are WT, temperature, and nutrient availability [18,19]. WT determines the depth of the oxic/anoxic boundary and controls the redox conditions [10,20,21]. Lowering of WT increases the supply of oxygen, and therefore, accelerates aerobic decomposition. This increases DOC production and changes its chemical composition and biodegradability [22,23]. However, in field conditions, DOC is subject to numerous dynamic processes. During dry periods, the DOC export is reduced, leading to DOC accumulation in pore water that is either consumed by microbes or leached to watercourses during a high runoff [22–25]. The microbial activity is strongly enhanced by higher temperature, which especially increases the release of recalcitrant DOC (Laurén et al., 2019). Additionally, the availability of nitrogen (N) and its ratio to C can limit organic matter decomposition processes [26–28].

Peatland drainage and forest harvesting modify multiple factors controlling the production, decomposition, and transport of DOC. Clear-cutting removes the evapotranspiration of the tree canopy, which leads to high WT and can increase DOC export from the peatland [29,30]. Drainage or ditch maintenance and the following increase in tree biomass lowers the WT, which accelerates the decomposition processes and results in a higher release of more recalcitrant DOC [31,32]. Besides increasing tree biomass and woody litter input, drainage alters litter quality and quantity also by inducing a succession of the ground vegetation towards graminoid and shrub dominance [33,34]. This can lead to a higher input of plant-derived and more labile DOC through increased root exudate production and leaching from the decomposing tree, shrub, and graminoid litter than from the *Sphagnum* litter or peat itself [24,30,35]. Clear-cutting sets off another vegetation succession with pioneer species, such as grasses and broad-leaved trees [36]. The pioneer species produce higher-quality substrates for microbes that do not support soil C accumulation [36–38].

Since drainage and clear-cutting have both long- and short-term effects on DOC and nutrient leaching with further ecological consequences [14,32,39], alternative management options are needed. Continuous cover forestry (CCF) has been suggested as an alternative for clear-cutting, which in field experiments has either decreased or increased the soil C and nutrient content compared to uncut forests [40–43]. The CCF methods include selective harvesting, avoiding soil preparation, and natural regeneration [39,44]; therefore, CCF may have less impact on soil C and nutrient levels than clear-cutting. Since CCF likely maintains low WT through high evapotranspiration, it is assumed to cause less DOC and nutrient leaching than clear-cutting [29,30,45,46]. These changes in runoff may abate the browning of inland waters [39]. Yet, the studies on the effects of the so-called "soft" forestry methods on DOC processes remain scarce [12,30,39].

To fill this knowledge gap, we set up a column experiment using peat from unharvested (control), CCF, and clear-cut drained peatland forests to compare the effects of different forest management practices on pore water DOC quality and concentration as well as the biodegradation potential of DOC to $CO_2$. This ex situ approach allows us to unify the abiotic environment between the treatments and to test the effect of WT on the pore water DOC without the background noise related to field-based studies. The outcomes provide urgently needed tools for science-based decisions towards C-neutral and "water-friendly" forest management. Additionally, to approach differences between natural and strongly modified peatlands forming a direct terrestrial–aquatic continuum, columns are also collected from an undrained alluvial sedge fen. By comparing the columns from these two sites with differing initial and current ecohydrology and vegetation, we can further discuss the role of vegetation and hydrology on pore water properties and DOC biodegradation. We monitor the pore water DOC and dissolved total nitrogen (DN) concentrations and spectral properties depicting DOC quality (i.e., $SUVA_{254}$, E2:E3, and E4:E6-ratios) and DOC biodegradation to $CO_2$.

We hypothesise that the pore water characteristics and biodegradability are more similar in waters collected from columns from the unharvested and CCF than the clear-cut drained peatland forest, and expect differences, especially increased lability, in the clear-cut site due to changes in the ground vegetation. We further hypothesise higher concentrations and more labile DOC in the pore water from the sedge fen than from the drained site with forest-like ground vegetation and peat exposed to drainage for eight decades. In all sites, we expect to see higher DOC lability and concentrations in high WT treatments due to the easier access of the DOC to water. Consequently, we expect to see higher $CO_2$ release in high WT treatments.

## 2. Materials and Methods

### 2.1. Soil Column Collection and Experiment Set-Up

The production of DOC and its biodegradation to $CO_2$ were studied in a laboratory experiment with peat columns collected from Paroninkorpi experimental site (61.01° N, 24.75° E) and Sudenpesänkangas nature conservation site (61.12° N, 25.11° E), both located in southern Finland. Paroninkorpi is a minerotrophic nutrient-rich drained peatland forest [47] where the main ditch network was dug in the 1940s and complemented at the beginning of the 1960s. The 0.6–0.7 m deep ditches are spaced at 65–70 m intervals. The site likely had a patchy forest prior to drainage, which had developed and formed a closed canopy Norway spruce (*Picea abies* (L.) Karst.) forest until the harvesting operations in 2017 [47]. The site was divided into experimental plots in February 2017. On the plots, three different management operations were conducted: (1) in the unharvested plots the basal area was 25 $m^2$ $ha^{-1}$ at the beginning of the study, (2) in the continuous cover forest (CCF) the basal area was 12 $m^2$ $ha^{-1}$, and (3) in the clear-cut plots all trees were removed (remaining basal area of 0 $m^2$ $ha^{-1}$) (see [30] for further details on the setup). For the CCF, selective cutting was conducted retaining the suppressed and understory trees and some of the largest trees. After the operations, the ground vegetation was formed by dwarf shrubs, mosses (mainly *Sphagnum* sp.), and ferns in the unharvested and CCF plots and of dense patches of raspberry (*Rubus idaeus* L.), birch (*Betula* sp.), and occasional ferns in the clear-cut plots. The peat depth was 1.5 m and formed of *Carex*-wood peat. At Sudenpesänkangas, the peat samples were collected from a treeless, alluvial sedge fen. The fen formed a 40-metre wide peatland strip between the nearby forest-covered mineral soil and a small brook. Thus, the sedge fen received nutrient-rich waters both through runoff from the nearby forest and flooding of the brook caused by snowmelt and a beaver occasionally operating in the upper stream. The peat layer thickness was more than 1 m, and the peat was composed of sedge, wood (*Betula* sp.), and reed remnants. The dominant vegetation was *Carex* sp. with *Polytrichum* sp. and *Sphagnum* sp. dominant in the moss layer.

Sixteen undisturbed soil columns were collected from each treatment (clear-cut, CCF, unharvested) in Paroninkorpi and 12 in Sudenpesänkangas in January–February 2020. Sam-

pling in the mid-winter, when the soil processes are the slowest, minimized the disturbance to the soil processes. The collection was done with industrial-grade PVC cylinders (15 cm in inner diameter and 50 cm in height). The ground vegetation was included in the columns. The columns were wrapped in plastic to avoid drying and kept in dark at +4 °C before starting the eight-month experiment in March 2020.

For the experiment, the columns were set in a greenhouse with a semi-light-permeable roof in a random order to minimize the possible position effects. Rhizon soil water samplers (pore size 0.6 μm; Rhizosphere Research Products, Wageningen, Netherlands) were installed horizontally into the columns at 5 cm from the bottom for water extraction. The lower ends of the columns were wrapped into polyethylene bags to prevent water leaking from the bottom. Finally, the columns were placed into 12-L buckets, which were randomly divided into two different WT treatments: low (WT at 40 cm below the peat surface) and high (WT at 20 cm below the peat surface). The low WT corresponded to the annual average WT in Paroninkorpi [30]. The WT was set in the beginning and maintained by watering from above weekly throughout the experiment with deionized water (pH adjusted with HCl to 5.5 that corresponds to the rainwater pH). The soil temperature (at 5 cm depth) was monitored at two-hour intervals during the experiment with Comet Multilogger M1321 (Comet Systems Ltd., Rožnov pod Radhoštěm, Czech Republic) from one of the columns.

## 2.2. Sampling and Analysis

The water collection from the columns started in April 2020, i.e., after a month of initial equilibration period. Five sets of pore water samples were collected during the experiment period (April, June, July, August, and October). The DOC biodegradability was studied right after sample extraction by incubating three-replicate 20 mL aliquots of the waters in 100 mL glass bottles. The experiments consisted of multiple 72 h incubation cycles. The first cycle started at the onset of the experiment (days 0–3, i.e., T3) and was repeated at 6–9 (T9), 18–21 (T21), and 28–31 (T31) days after the onset. In April, the incubations were carried out also for days 62–65 (T65) to test how long the biodegradation continued. Since the degradation was low at T65, as a compromise, in July, August, and October the last incubation was done for days 38–41 (T41). At the beginning of each incubation cycle, the bottles were capped outdoors with septa and aluminium screw caps, and 20 mL of outdoor air was injected into the bottles. The bottles were shaken in an orbital shaker (3 min, 80–100 rpm), after which 25 mL gas samples were taken from the headspace of each bottle with a syringe and a needle and injected into pre-evacuated 12 mL Exetainer vials (Labco, Lampeter, UK). The bottles were placed into a 15 °C incubator for 72 h, after which the bottles were shaken again and sampled for gases similar to at the beginning of the incubation cycle. Between the incubation cycles, the bottles were sealed with perforated aluminium foil and stored in the incubator at 15 °C.

The gas samples were analysed with an Agilent Gas Chromatograph (GC 7890A Agilent Technologies, Palo Alto, CA, USA) and an 80/100 Agilent Hayesep Q (6 ft × 1/8″ × 2 mm) column. The analyses were performed with a flame ionisation detector (FID) using helium as a carrier gas and synthetic air (450 mL min$^{-1}$) and hydrogen (40 mL min$^{-1}$) as flame gases. In addition, nitrogen gas (5 mL min$^{-1}$) was used as the make-up gas for the FID standard. The oven temperature was set to 60 °C with the detector temperature being 300 °C. The $CO_2$ concentrations were measured using a 4-point calibration curve determined with 433, 750, 1067, and 1500 ppm standard gas concentrations (Oy AGA Ab, Espoo, Finland).

In addition to the biodegradation experiment, DOC and DN concentrations were determined from the water samples with a Jena Multi N/C® 2100 analyser (Analytik Jena AG, Jena, Germany). The carbon to nitrogen (C:N) ratios were calculated. The quality (aromaticity) of DOC was investigated with a Shimadzu UV-1800 UV-VIS spectrophotometer (Shimadzu, Kyoto, Japan) using a 1 cm quartz cuvette. The specific ultraviolet absorbance at 254 nm (SUVA$_{254}$; L mg$^{-1}$ m$^{-1}$) was calculated by normalising UV absorbance at 254 nm to DOC concentration and the E2:E3- and E4:E6-ratios by dividing the absorbance at 250 nm (E2) with the absorbance at 365 nm (E3) and the absorbance at 465 nm (E4) with the ab-

sorbance at 665 nm (E6), respectively. The $SUVA_{254}$ correlates positively with complex macromolecular DOC rich in aromatics [48], and E2:E3 correlates inversely to DOC aromaticity and molecular weight [49]. The E4:E6-ratios depict DOC quality and are often used as a measure of humification with low values indicating the dominance of aromatic constituents [50]. The water for DOC, DN, and spectroscopy was always taken a day before the water for the DOC biodegradability incubations and stored at $-21\,°C$ before analysis. The samples for spectral analysis were diluted so that their absorbance values were <1.5.

Peat properties were determined from the samples collected from the topmost 50 cm of peat in the field; we analysed pH (mixed peat and deionized water; 1:2, *V:V*), electrical conductivity, dry bulk density (g cm$^{-3}$; dried for 48 h in 105 °C), and peat decomposition stage using von Post method [51]. From the samples collected at the end of the experiment, total concentrations of C and N were determined (Elementar Vario Max, Hanau, Germany). The photosynthesizing biomass per plant functional group (*Sphagnum* mosses, sedges, herbs, trees, graminoids, others) was determined from each column at the end of the experiment (dried at 40 °C until constant weight).

### 2.3. Calculations

In the DOC biodegradation experiments, the solubility of $CO_2$ in water *(KH(T))* as mol kg$^{-1}$ bar$^{-1}$ was calculated according to the Henry's law as follows:

$$K_H(T) = K_H^{\circ}\, e^{[C \cdot (\frac{1}{T} - \frac{1}{298.15})]} \tag{1}$$

where $K_H^{\circ}$ is Henry's law constant (0.034 mol kg$^{-1}$ bar$^{-1}$) for solubility for $CO_2$ in water at 298.15 K (mol kg$^{-1}$ bar$^{-1}$), *C* is temperature dependence constant (2400), and *T* is water temperature (K) in bottles.

The amount of $CO_2$ dissolved in water ($CO_{2W}$ mol L$^{-1}$) was calculated as follows:

$$CO_{2W} = (K_H\,(T) \cdot 10^{-5}) \cdot \left(PCO_2 \cdot 10^{-6}\right) \cdot P_{atm} \cdot V \tag{2}$$

where *KH(T)* is the solubility of $CO_2$ in water converted to mol L$^{-1}$ Pa$^{-1}$, $PCO_2$ is the measured $CO_2$ concentration as ppm converted to percentage, $P_{atm}$ is the gas partial pressure under 1 atm ($1.0135 \times 10^5$) above the boundary of a solution, and *V* is the water volume (L) in the bottle.

The amount of $CO_2$ released in the air space of the bottle ($CO_{2A}$ mol L$^{-1}$) was calculated based on the ideal gas law as follows:

$$CO_{2A} = \frac{0.001 \cdot P_{atm} \cdot (PCO_2 \cdot 10^{-6}) \cdot V}{R \cdot T} \tag{3}$$

where 0.001 is the coefficient for converting gas volume from m$^3$ into L, $P_{atm}$ is the partial pressure of the $CO_2$ under 1 atm ($1.0135 \times 10^5$), $PCO_2$ is the measured $CO_2$ concentration as ppm converted to percentage ($PCO_2 \times 10^{-6}$), *V* is the air volume (L) in the bottle, *R* is the universal gas constant (8.3145 J mol$^{-1}$ K$^{-1}$), and *T* is the temperature (K).

The $CO_2$ release rate (mol L$^{-1}$ day$^{-1}$) in the bottle during the incubations is the combination of $CO_2$ dissolved in water and released in the air:

$$CO_2 = CO_{2W} + CO_{2A} \tag{4}$$

For estimating the daily $CO_2$ release, the results were divided by the incubation time, i.e., three days. The $CO_2$ production per DOC was calculated by dividing the $CO_2$-C emitted per day by the DOC concentration.

### 2.4. Statistics

The normality and homogeneity of the data were checked visually, and outliers were then identified as values 1.5 times the interquartile range above the third quarter or below

the first quarter. As a result, a total of 43 outliers (out of 300 values) were detected and removed only in the $SUVA_{254}$ data. Analysis of variance (ANOVA) was used to explore the differences in the $CO_2$ production at the DOC biodegradation experiment as well as in the initial DOC and DN concentrations and DOC quality parameters ($SUVA_{254}$, E2:E3- and E4:E6-ratio). General linear model (GLM) was used to test interactions of factors (month, WT, and management). Tukey's test was used as the post hoc analysis. Levene's test was used to assess the homogeneity of variances. The significance level ($p$) for the statistical tests was set at 0.05. Additionally, an ordination method was used to summarise the relationships between the measured parameters. In a redundancy analysis (RDA), the $CO_2$ production (during T3 as most of the $CO_2$ was produced on the first incubation) and the measured pore water DOC and DN concentrations were used as response variables, while peat parameters (peat temperature at 5 cm depth, and peat C, N, dry bulk density), pore water quality (concentrations of DOC and DN, $SUVA_{245}$, E2:E3, and E4:E6-ratios), and vegetation biomass (biomass measured at the end of the experiment) were used as explanatory variables. The site, interaction factor of forest management and WT treatment, and the month were used as supplementary variables. The RDA analysis was conducted with 999 Monte Carlo permutations. The statistical analyses were done with IBM SPSS Statistics software, version 26 (SPSS Inc., Chicago, IL, USA) and ordination analyses with Canoco5 for Windows (Microcomputer Power, Ithaca, NY, USA).

## 3. Results

### 3.1. Biodegradation

Throughout the incubation experiment, most of the $CO_2$ produced by DOC biodegradation took place during the first 72 h incubation period (T3; $p < 0.05$; Figures 1 and S1). Additionally, WT affected DOC biodegradation at the beginning of the incubation rounds (April T9 $p = 0.04$, June T3 $p = 0.02$, July T3 $p = 0.03$, October T9 $p = 0.04$) and despite statistical differences were otherwise barely detected between the WT treatments due to high deviations, both the daily $CO_2$ production per volume ($\mu g\ CO_2\ L^{-1}\ day^{-1}$) and per DOC ($\mu g\ CO_2$-C $DOC^{-1}\ d^{-1}$) were generally higher in the pore waters collected from the high WT treatments.

The forest management had little or no effect on DOC biodegradability (Figure 1). In general, the daily $CO_2$ production per volume ($\mu g\ CO_2\ L^{-1}\ day^{-1}$) was lowest in the sedge fen (July and August $p < 0.05$, other months $p > 0.05$; Figures 1a and S1). Timewise, the lowest $CO_2$ production per volume at T3 was detected in April ($p < 0.05$ in all but the sedge fen), when it varied between 30 and 116 $\mu g\ L^{-1}\ day^{-1}$. The highest range was in July between 25 and 316 $\mu g\ L^{-1}\ day^{-1}$. (Figure 1a). From T9 onwards the $CO_2$ production was low (below 18 $\mu g\ L^{-1}\ day^{-1}$) in April, June, and July but higher in August and October (up to 100 $\mu g\ L^{-1}\ day^{-1}$ and 65 $\mu g\ L^{-1}\ day^{-1}$, respectively).

The daily $CO_2$ production in relation to DOC content ($CO_2$ per DOC; $\mu g\ CO_2$-C $DOC^{-1}\ d^{-1}$) was higher in the sedge fen than in the forestry drained peatland treatments in high WT in April ($p < 0.05$; on T9 and T31) and in low WT in October ($p < 0.05$; on T21, T31, and T41; Figures 1b–f and S2). There were no statistical differences in pore water DOC biodegradation between the clear-cut and unharvested forest (Figure 1b–f). However, in July on T3, the high WT in clear-cut had higher $CO_2$ per DOC than the low WT in CCF ($p < 0.05$).

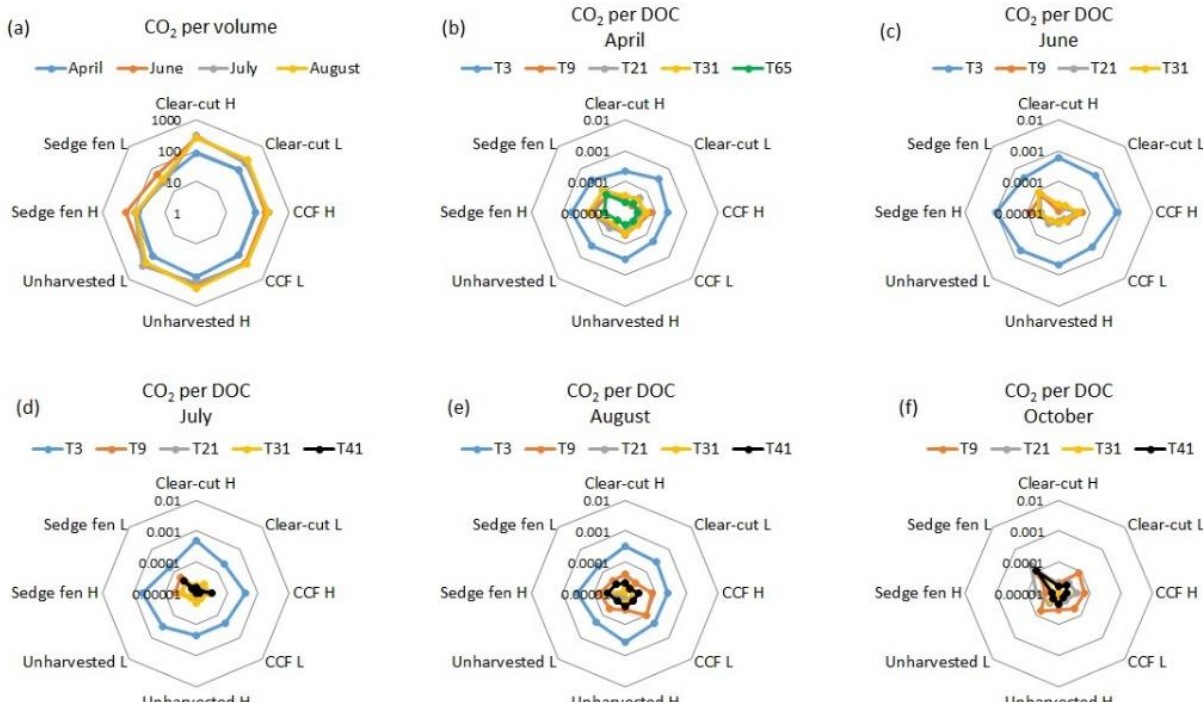

**Figure 1.** (**a**) Daily $CO_2$ production per water volume (µg $CO_2$ $L^{-1}$ $day^{-1}$) at T3 in April, June, July, and August in the water extracted from peat columns representing clear-cut, continuous cover forestry (CCF; 12 $m^2$), and unharvested forest in Paroninkorpi and alluvial sedge fen in Sudenpesänkangas. The $CO_2$ production expressed per DOC content (µg $CO_2$-C $DOC^{-1}$ $d^{-1}$) during the same incubations in (**b**) April, (**c**) June, (**d**) July, (**e**) August, and (**f**) October. The incubations consisted of multiple 72 h-long cycles. The first cycle started at the onset of the incubation (days 0–3, i.e., T3) and was repeated at 6–9 (T9), 18–21 (T21), and 28–31 (T31) days after the onset. In April the incubations were done also for days 62–65 (T65), and in July, August, and October for days 38–41 (T41). H = high water table, L = low water table; Nb. logarithmic scale. The T3 samples in October were accidentally destroyed during analysis.

### 3.2. Pore Water Quality

In general, the average pore water DOC concentrations varied from 43 to 209 mg $L^{-1}$ and increased during the experiment ($p < 0.05$; Figures 2a and S3). The lowest DOC concentrations were detected in the sedge fen columns ($p < 0.05$), whereas there were practically no differences in the DOC concentrations between the clear-cut, CCF, and unharvested drained forest (Figure 2a). The DN concentrations increased during the experiment and were, in general, lowest (<5 mg $L^{-1}$) in the sedge fen (Figure 2b). In August, the DN concentrations were higher in CCF and unharvested forest than in clear-cut plots ($p < 0.05$; Figure S3) but otherwise the DN concentrations did not differ between the forest plots. The pore water C:N-ratios varied between 12 and 30, being higher in the beginning than at the end of the experiment and generally lowest in the sedge fen (Figures 2c and S3). The effect of WT on DOC and DN concentrations and C:N ratios was marginal. The low WT led to higher DOC concentration than high WT at the beginning of the experiment in CCF and in clear-cut (in April $p < 0.05$) and lower DOC and DN in CCF in July ($p < 0.05$). The clear-cut had the highest C:N-ratios in both high and low WT at the beginning of the experiment (in April $p < 0.05$) and in low WT also in June and July ($p < 0.05$).

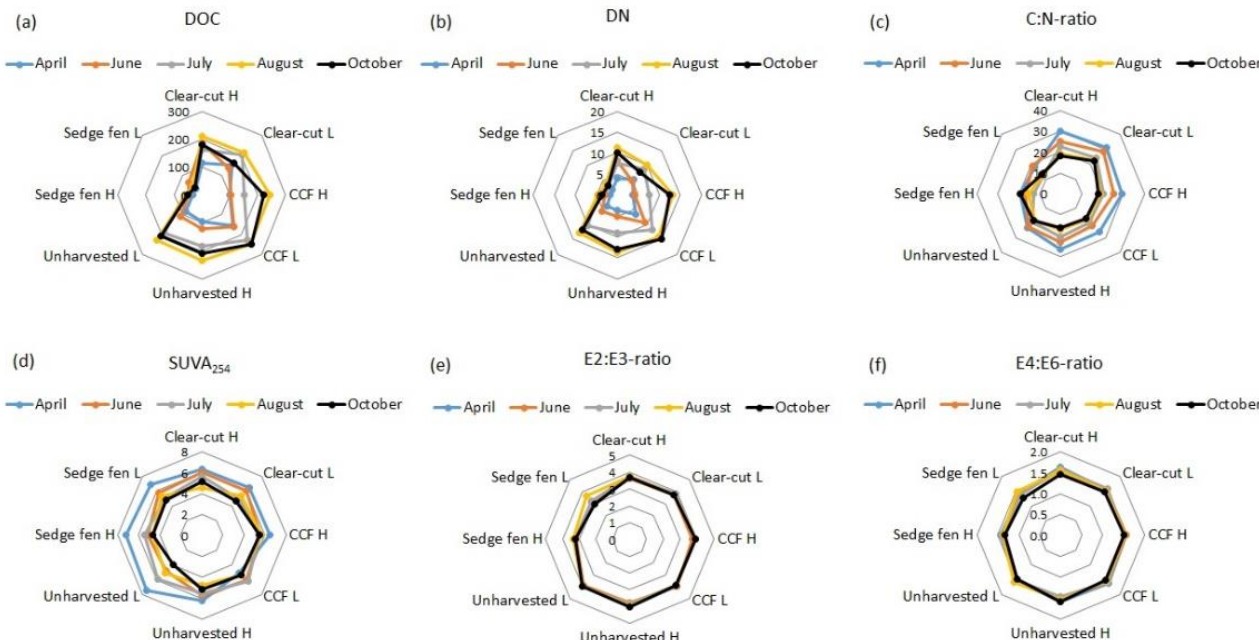

**Figure 2.** Water (**a**) dissolved organic carbon (DOC; mg L$^{-1}$) and (**b**) dissolved nitrogen (DN; mg L$^{-1}$) concentrations, (**c**) carbon to nitrogen (C:N) -ratios, (**d**) SUVA$_{254}$ (L mg$^{-1}$ m$^{-1}$), and (**e**) E2:E3 and (**f**) E4:E6-ratios in the peat columns collected from the clear-cut, continuous cover forestry (CCF; 12 m$^2$), unharvested forest in Paroninkorpi, and alluvial sedge fen in Sudenpesänkangas. H = high water table, L = low water table.

The aromaticity of DOC was estimated with SUVA$_{254}$ and E2:E3 and E4:E6-ratios. The SUVA$_{254}$ values (range 4.4–7.8) did not differ between the columns, whereas the E2:E3 (range 3.0–4.0) and the E4:E6-ratios (range 1.3–1.6) were lower in the columns collected from the sedge fen than from the drained peatland (clear-cut, CCF, unharvested), especially in the first three water samplings (April, June, and July; $p < 0.05$; Figure 2, Table S1). The WT had no effect on SUVA$_{254}$ or the E2:E3 and E4:E6-ratios. However, the pore water SUVA$_{254}$ values were in general highest in April ($p < 0.05$) and lowered towards August (Figure 2, Table S1), whereas there were no time-related changes in the E2:E3 and E4:E6-ratios.

### 3.3. Peat Properties and Biomass on Columns

The peat dry bulk density and pH were highest in the sedge fen and the clear-cut plots (Table S2). The electrical conductivity was lowest in the clear-cuts and highest in the unharvested forest and at the sedge fen. At the end of the experiment, the peat C concentration was 43.5–53.9% (gravimetric), being highest in the sedge fen. The peat N concentration was 1.60–2.50% of the dry mass, being on average highest in the uncut plots. Peat C:N-ratio was highest in the sedge fen (on average 28.8) and in the three drained managements between 22.6 and 25.1.

The average dry photosynthesizing biomass on the columns was highest in the sedge fen and clear-cut forest (6.5 g and 6.5 g, respectively, corresponding to 3700 kg ha$^{-1}$) and lowest in the forest-covered plots (CCF 3.4 g, unharvested 2.9 g, corresponding to 1600–1900 kg ha$^{-1}$; Table S3). In the sedge fen, the biomass was mainly formed of sedges and other graminoids (89.3 ± 28.6% and 8.33 ± 28.9%, respectively) and there were no *Sphagnum* mosses growing in the columns. In the clear-cut and forest-covered drained peatlands, the dry biomass was mainly formed of a combination of *Sphagnum*, sedge or grasses, other graminoids, and tree leaves (20.5–45.1%, 0–33.8%, 15.4–27.4%, and 7.13 ± 9.21%, respectively).

*3.4. Summarising the Patterns Describing Pore Water Quality*

In the redundancy analysis (Figure 3), the explanatory variables explained a total of 84.4% (adjusted 78.1%) of the variation in the response variables. The strongest Axis-1 (summarises 74.1% of the variation) separated DOC, DN, and $CO_2$ per volume from $CO_2$ per DOC. Of the explanatory variables, the E2:E3 and E4:E6-ratios aligned with the $CO_2$ per volume and pore water DOC, DN, and CN concentration. Of the supplementary variables, the forestry drained sites aligned with the pore water E2:E3 and E4:E6-ratios and $CO_2$ per volume. Additionally, Axis1 separated the sedge fen, which aligned with the peat properties and biomass, from the forestry drained sites, which aligned with the $CO_2$ per volume and pore water DOC, DN, and CN concentration and the E2:E3 and E4:E6-ratios. On the Axis2 (explains 7.3%), the $CO_2$ per DOC aligned with $SUVA_{254}$, while the pore water C:N-ratio and peat N concentrations pointed to contrary directions. Most of the high WT treatments, as well as April and June of the measurement months, aligned with the $SUVA_{254}$ of explanatory variables and with the $CO_2$ per DOC and pore water DN concentration of response variables, Most of the low WT treatments as well as August and July aligned with the peat N concentration of the explanatory variables.

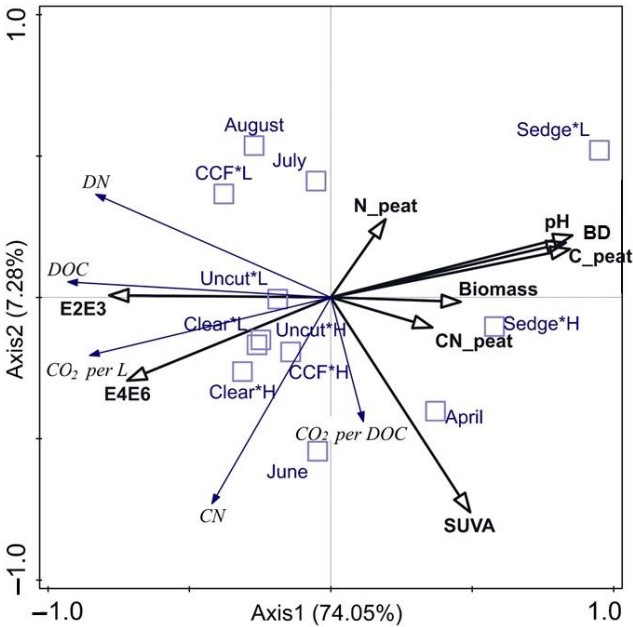

**Figure 3.** RDA-all-in. A constrained analysis using $CO_2$ data (expressed as $CO_2$ per L and $CO_2$ per DOC), and the measured pore water C and concentrations are used as response variables (italics; blue arrows). The peat properties and pore water spectral data were used as explanatory variables (bold; black arrows). Month and the interaction factor of management and water table treatment were used as supplementary variables (blue boxes). $CO_2$ per L = $CO_2$ production per volume ($\mu$g $CO_2$ $L^{-1}$ $day^{-1}$) on the first three days of incubations (T3); $CO_2$ per DOC = $CO_2$ production per measured DOC ($\mu$g $CO_2$-C $DOC^{-1}$ $d^{-1}$) on the first three days of incubations (T3); C_peat = C concentration of dry peat; N_peat = N concentration of dry peat; CN_peat = peat C:N-ratio; BD = peat dry bulk density (g $cm^{-3}$); Biomass = total photosynthesizing aboveground dry biomass (g); L = low water table; H = high water table; clear = clear-cut; CCF = continuous cover forest.

## 4. Discussion

We studied pore water quality in terms of DOC and DN concentrations, spectral data, and DOC biodegradation on differently managed drained peatland forests (clear-cut, CCF, unharvested), as well as on an alluvial sedge fen. Additionally, we explored the effect of WT on pore water quality and DOC biodegradation rate. In contrast to our hypothesis, in general we found no differences in the concentration or quality of DOC, concentration of DN, or in the biodegradation rate of DOC between the CCF, clear-cut, and unharvested

forests. Unexpectedly, the pore water DOC concentrations were up to seven times lower and the DOC biodegradation rate per volume of water was up to three times lower in the sedge fen than in the drained peatland forests. However, the DOC biodegradation rate per DOC amount was highest in the sedge fen with high WT, supporting our assumption about higher biodegradability in the sedge fen. Yet, based on the spectral data, the pore water had higher aromaticity and MW in the sedge fen than in the drained peatland forests. As expected, the DOC biodegradation rate was higher in the high WT treatments than in the low WT treatments, but the WT had only a minor effect on DOC concentrations and no effect on $SUVA_{254}$ or on the E2:E3 and E4:E6. Some variation during the experiment was observed both in the pore water quality as well as in the $CO_2$ release per volume.

In general, we did not detect differences in the DOC biodegradation rate between the forest management practices. However, in a field study conducted at the same site, management had an effect on the groundwater quality and biodegradation [30]. The observed biodegradation rates (per volume and per DOC) were, in general, similar to in the field study and in both studies most of the $CO_2$ was released within the first three incubation days (T3). Palviainen et al. (2021) found higher DOC and DN and lower C:N-ratios in groundwater collected from the clear-cut sites than from the CCF and unharvested controls and attributed the differences to the higher WTs, changes in vegetation, and elevated temperatures accelerating decomposition in the clear-cut plots. The lack of management effect in our study is likely attributed to controlled abiotic conditions (WT, temperature, lack of trees) between the peat columns from differently managed plots. Thus, it can be interpreted that peat being exposed to long-term drainage is not vulnerable to changes per se, but that forest management alters multiple biotic and abiotic factors that independently and interactively control the production, transport, and decomposition of DOC.

While it is generally known that WT and water movement are important factors controlling DOC processes and transportation (e.g., [52–54]), we detected a clear WT effect only on pore water $CO_2$ per DOC release. The observed higher pore water $CO_2$ per DOC release in high WT treatments was likely because the plant exudates and degradation of less decomposed surface peat provided input of labile DOC and because there was an enhanced hydraulic connection, i.e., a path for the produced DOC to the saturated water, due to the high WT. However, we did not detect differences in DOC concentration or quality between the WT treatments within the sites. This reflects opposite effects on the on-site WT between the sites: in the columns from the drained forest, the high WT treatment mimicked a rise in WT with respect to typical field conditions. Contradictorily, both of the WT treatments in the sedge fen samples mimicked the lowering of the WT because on sedge fens the WT is typically close to the surface (higher than −10 cm from the surface) and these kinds of sites are frequently flooded [55,56]. Such WT lowering can occur, for example, during summer droughts, which are expected to become more frequent in the future in boreal regions [57]. While it was assumed that in the sedge fen, exposing the initially waterlogged peat to aerobic decomposition would lead to the highest C release in pore water and higher biodegradability of DOC, the low WT columns had the overall lowest measured pore water DOC concentrations and $CO_2$ production. Therefore, it is likely that (i) in the drained sites, the high WT reduced the decomposition of peat, and that (ii) in the sedge fen, the low WT resulted in insufficient hydraulic connection between the surface and bottom peat layer and the DOC was retained in pore spaces (e.g., absorption, redox) or consumed by microbes [58,59]. However, the recalcitrant DOC formed during the low WT and reaching the groundwater forms a constant outflow of DOC [60], and the DOC formed in the upper parts can be flushed out from the peat once the WT rises, for example, as result of heavy rain [22]. Our findings suggest that while the high WT can increase the risk of leaching of labile DOC from drained peatlands, it likely suppresses decomposition processes and as a result leaching of the recalcitrant DOC.

The assumption of higher biodegradability of DOC in the sedge fen was supported by the higher $CO_2$ production per DOC in the high WT sedge fen columns. However, we detected lower pore water DOC concentrations and higher recalcitrance (high $SUVA_{254}$

and low E2:E3) in the sedge fen columns than in the drained site. The SUVA$_{254}$ values correlate positively with polyphenols and other phenolic compounds [61], which can occur together with labile DOC during the high WT [25,62]. Thus, the contradiction between the observed DOC biodegradation and the quality parameters may result from the fact that in addition to the labile fraction, the pore water consists of some polyphenols or other phenolic compounds that correlate positively with SUVA$_{254}$. However, this is against the generally reported negative correlation of SUVA$_{254}$ with the CO$_2$ production of DOC (e.g., [30,63]). Since our spectral data were determined from water samples that were frozen after sampling and thawed prior to spectral analysis, and especially since the UV absorbances related to labile C are prone to, for example, freezing and thawing [18,64,65], we cannot rule out the possibility that the results related to labile fractions (especially on the SUVA-values that likely caused the high number of noticed outliers), or more the lack of them, are an artefact of sample processing. Yet, this does not change the interpretation of the results, only the postulation of the causes of the outcomes.

The development of vegetation and progression of decomposition caused variation during the experiment in DOC biodegradability and quality primarily on the high WT columns. The CO$_2$ production per DOC peaked in the columns from the sedge fen, clear-cut, and CCF plots in June and July collaterally with the known peak of photosynthesizing biomass in boreal regions. This further links the higher biodegradability of DOC to plant origin and especially to the presence of vascular plants [24,35,66], which was more present and had higher biomass in the columns from the sedge fen and clear-cut plots. Contradictorily, the detected prolonged production (high fluxes after T3) of CO$_2$ per volume in the autumn and the increase in DOC and DN concentrations during the experiment in the columns from the drained study site indicate higher recalcitrance of the released DOC, and these are outcomes of soil decomposition processes [30,59,67]. Considering the longer lifetime of the recalcitrant DOC released in decomposition processes [15], these findings indicate that there is a higher risk of export of DOC and DN from drained sites; they are likely to end up in aquatic ecosystems with the runoff waters.

## 5. Conclusions

The three outcomes of this study and how they relate to forest management in drained peatland forests are:

1.  Peat, per se, does not cause detectable differences in DOC production or biodegradability between different forest management operations in peatland forests drained decades ago.
2.  Ground vegetation can be a relevant source of DOC [24,35,68]. Our results indicate that the biodegradability of DOC in pore water is linked to high vegetation biomass, for example, in primary succession after clear-cutting. Therefore, the effects of the ground vegetation and its succession, and logging residues that are known to increase soil C pools and DOC concentrations in pore water [30,69], should be studied as potential sources of easily biodegradable C into watercourses. With time, the ground vegetation succession and decomposition processes affect the properties of old peat and formation of new peat [33], which may drive peat properties differently depending on the management.
3.  High WT can reduce decomposition and formation of recalcitrant DOC, but it can also provide a pathway especially for labile DOC to groundwater.

**Supplementary Materials:** The following supporting information can be downloaded at: https://www.mdpi.com/article/10.3390/f13040599/s1, Figure S1: Daily CO$_2$ production per volume; Figure S2: Daily CO$_2$ production per DOC; Figure S3: Dissolved organic carbon and dissolved nitrogen concentrations and carbon to nitrogen -ratios in the peat columns; Table S1: SUVA$_{254}$ and E2:E3 and E4:E6-ratios in the peat columns; Table S2: Peat properties; Table S3: The average photosynthesizing biomass on the columns.

**Author Contributions:** Conceptualization, E.P., M.K., M.P., A.L., A.O. and J.P.; methodology, E.P., M.K., M.P., A.L., A.O. and J.P.; software, E.P. and M.K.; validation, E.P. and M.K.; formal analysis, E.P. and M.K.; investigation, X.Z., N.K. and H.A.; resources, A.O. and J.P.; data curation, E.P., M.K., X.Z. and N.K.; writing—original draft preparation, E.P. and M.K.; writing—review and editing, E.P., M.K., M.P., A.L., X.Z., N.K., H.A., A.O. and J.P.; visualization, E.P. and M.K.; supervision, M.P., A.L., A.O. and J.P.; project administration, M.P., A.O. and J.P.; funding acquisition, M.P., A.L., A.O. and J.P. All authors have read and agreed to the published version of the manuscript.

**Funding:** This research was funded by the Academy of Finland, project REFORMWATER (326818) and CASCAS (323997, 323998), and the Ministry of Agriculture and Forestry, project Kokonaiskestävää puuntuotantoa turvemailta—SUO. Open access funding provided by University of Helsinki.

**Institutional Review Board Statement:** Not applicable.

**Informed Consent Statement:** Not applicable.

**Data Availability Statement:** Data is available from the authors on request.

**Conflicts of Interest:** The authors declare no conflict of interest. The funders had no role in the design of the study; in the collection, analyses, or interpretation of data; in the writing of the manuscript; or in the decision to publish the results.

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
