# Peer review of "Impact of Forest Harvesting Intensity and Water Table on Biodegradability of Dissolved Organic Carbon in Boreal Peat in an Incubation Experiment"

_forests, doi:10.3390/f13040599_

Round 1

Reviewer 1 Report

The manuscript “Impact of forest harvesting intensity and water table on biodegradability of dissolved organic carbon in boreal peat in an incubation experiment" by Peltomaa et al. fills an important knowledge gap on the ecology of northern peatlands. Currently, data on DOC dynamics is lacking on these systems, which limits our ability to make informed predictions on their carbon cycling trajectory under increasing land use pressures and a changing climate. This paper presents important insights into mechanisms that control DOC consumption and production. The real strength of this study is the use of sites that have experienced long-term drainage because it provides valuable data for modeling; therefore, providing better predictions of future changes to these systems. The results are not directly relatable to field condition, but nonetheless I feel these types of experiments are extremely valuable to further our understanding of wetland processes. Overall the paper was well written. The questions were testable with the experimental design and provided insights into water table and temperature changes and substrate quality (different peatland plant types) on DOC production. After the following very minor issues are addressed, then this paper will be suitable for publication and I envision that it will be a highly useful resource, especially to the peatland community. Therefore, I highly recommend the publication of this paper in Forests.

Line 134 – 145 – It would be extremely informative to list the species of Sphagna and sedges because of differences in their tissue chemistry that can affect production/consumption processes in peatlands.

Line 149-150 – Indicate the amount of compression that took place when extracting the cores. It is very difficult to obtain a core in this manner without compressing the peat and it is useful to know how this affected the bulk density and could be different from the bulk density you measured in the field – Line 212.

Line 186-193 – Please state the GC column specs

Line 244 – “a total of 43 outliers (out of 300 values) were detected and 244 removed only in the SUVA254 data”  This is a lot of data points to remove (~14%). Are you sure these data did not provide insights into some undetermined relationship or process? Were you able to perform a multivariate analysis that included these data to see if they group together and/or have a relationship to any of your measured environmental variables?

Line 207 – “and 207 stored at -21 ℃ before analysis.” Please state how long these were stored because extended storage can affect spectra measurements.

Line 435 – Change citation format to match rest of the paper

Line 476-477 - differently “at” depending ?

Author Response

  • Line 134 – 145 – It would be extremely informative to list the species of Sphagna and sedges because of differences in their tissue chemistry that can affect production/consumption processes in peatlands.

Thank you for this comment; we totally agree that it would have been informative to have the species list. However, unfortunately, the plants were not identified on the species level, but functional level only.

  • Line 149-150 – Indicate the amount of compression that took place when extracting the cores. It is very difficult to obtain a core in this manner without compressing the peat and it is useful to know how this affected the bulk density and could be different from the bulk density you measured in the field – Line 212.

Thank you for the comment; unfortunately, we do not have the comparisons on the bulk density data as the soils from the columns have been stored for further microbial analysis and thus, we have not been able to measure the BD yet. However, the samples were collected in the winter when the surface soil was already frosted. Thus, the effect of the compression on the peat columns can be expected to be negligible or on a similar level as the disturbance caused by collecting the bulk density samples with an auger for the field samples used to measure the BD.

  • Line 186-193 – Please state the GC column specs

This information is now added on line 188.

  • Line 244 – “a total of 43 outliers (out of 300 values) were detected and 244 removed only in the SUVA254 data”  This is a lot of data points to remove (~14%). Are you sure these data did not provide insights into some undetermined relationship or process? Were you able to perform a multivariate analysis that included these data to see if they group together and/or have a relationship to any of your measured environmental variables?

Thank you for the comment; most outliers as described in the manuscript were greater than 10, i.e. clearly erroneous. We recognize that the freezing of samples could have an effect on spectral data and thus also on SUVA. Please, see our response on the next comment as well as our addition on lines 460-466.

  • Line 207 – “and 207 stored at -21 ℃ before analysis.” Please state how long these were stored because extended storage can affect spectra measurements.

Thank you for the comment; the samples were stored for one to two months. We recognize this may have affected the spectral measurements and thus we added the following text on lines 460-466: Since our spectral data was determined from water samples that were frozen after sampling and thawed prior to spectral analysis, and since especially the UV absorbances related to labile C are prone for example freezing and thawing [18,64,65], we cannot rule out the possibility that the results related to labile fractions (especially on the SUVA-values that likely caused the high number of noticed outliers), or more on the lack of them, is an artifact of sample processing. Yet, this does not change the interpretation of the results, just the postulation of the causes of the outcomes.

  • Line 435 – Change citation format to match rest of the paper

This is now fixed.

  • Line 476-477 - differently “at” depending ?

“at” is now removed.

Reviewer 2 Report

The paper is clear and well written, I have only some minor remarks.

Introduction provides good review of literature, I suggest only to add some short characterization (or definition) of CCF management.

Methods:

I suggest to add some schematic map with research plots.

Could be the term "nutrient-rich" (l 124) specified or quantified?

Results:

l 303: I would prefer values of DOC concentrations, instead of p<0.05. Or maybe begin with the second sentence (l 305), and then continue with originally the first. This could help (at least me) to follow your message.

l 335-337: Give some values, please, not only their comparison.

Discussion:

l 399 "biodegradation rate was (per volume and per DOC) were..."

Author Response

  • Introduction provides good review of literature, I suggest only to add some short characterization (or definition) of CCF management.

We are a bit confused by this comment; CCF is already described in the introduction on lines 86-94. Hopefully this answers to the Reviewer’s comment.

Methods:

  • I suggest to add some schematic map with research plots.

The research area has been described more in detail in Palviainen et al. 2021 (reference [30]), where also a map of the setup is presented. We added this information on line 133.

  • Could be the term "nutrient-rich" (l 124) specified or quantified?

We have added the following text and a reference on line 124: “Paroninkorpi is a minerotrophic nutrient-rich drained peatland forest [47],”

Results:

  • l 303: I would prefer values of DOC concentrations, instead of p<0.05. Or maybe begin with the second sentence (l 305), and then continue with originally the first. This could help (at least me) to follow your message.

These two sentences have now been re-organized (lines 309-314).

  • l 335-337: Give some values, please, not only their comparison.

The values have now been added on lines 335 and 336.

Discussion:

  • l 399 "biodegradation rate was (per volume and per DOC) were..."

This is now fixed.